# When winning costs your peace: How does vertical individualism Hijack relaxation capacity? Network analysis and mediation models

**Wenyan Cheng**‡, **Ling Cheng**‡, **Zhichao Qian**, **Haoyu Wang***

College of Education Science, Anqing Normal University, Anqing, China

‡ Wenyan Cheng and Ling Cheng are co-first authors for this study.
* wanghaoyu904045570@126.com

## Abstract

In the context of increasing competition, the phenomenon of individuals experiencing guilt or anxiety at rest has become more pronounced, particularly among Chinese university students. While previous research has primarily explained this phenomenon from the perspective of collectivist cultures, this study posits that vertical individualism may offer a more compelling explanation. A sample of 550 Chinese university students was surveyed to collect data on vertical/horizontal individualism-collectivism, status anxiety, and rest intolerance. A partial correlation network analysis, controlling for demographic covariates, was conducted to explore the psychological structure of these constructs. The results identified Vertical Individualism (VI) and Status Anxiety (SA) as the core bridge nodes connecting the community of cultural values to the dimensions of rest intolerance. Subsequent mediation analyses confirmed that SA partially mediated the relationship between VI and overall rest intolerance. This indirect effect was particularly pronounced for the affective and social-comparative components of the phenomenon. These findings challenge traditional collectivist frameworks and reveal a nuanced psychological mechanism: competitive cultural values exacerbate rest intolerance through the pathway of status anxiety. This provides novel theoretical insights for psychological interventions and cultural adaptation education in higher education settings.

## 1. Introduction

In the accelerated rhythm of contemporary society, many people fail to experience genuine relaxation during leisure time, instead encountering negative emotions, such as guilt and anxiety. Although sufficient leisure time contributes positively to psychological and physical well-being [1], the interplay of sociocultural factors and internalized belief systems frequently prevents university students from fully engaging in and enjoying leisure activities without cognitive interference. This phenomenon, termed

**Data availability statement:** The relevant data are available in a public repository at https://doi.org/10.57760/sciencedb.psych.00638.

**Funding:** This research was funded by the Key Programs of Humanity and Social Science Research Project of Anhui Educational Committee, 2024AH052762.

**Competing interests:** The authors have declared that no competing interests exist.

"rest intolerance," refers to feelings of guilt, anxiety, or discomfort that individuals experience during periods of rest or recreational pursuit [2]. Such negative affective responses during leisure time represent a significant psychological construct worthy of empirical investigation, given their potential implications for mental health outcomes in achievement-oriented cultural contexts.

The etiology of rest intolerance is influenced by multiple factors, with cultural background significantly affecting individuals' perceptions of rest and leisure activities. Research indicates that collectivistic cultural orientations emphasize social responsibility and diligent work [3], potentially intensifying perceived pressure and self-reproach during leisure engagement. Conversely, individualistic cultures prioritize personal choices and fulfillment, which may result in diminished guilt during periods of leisure [4]. Furthermore, status anxiety has been identified as another critical determinant, referring to individuals' concerns about missing advancement opportunities or diminishing social standing through perceived "unproductive" time use [5,6]. These apprehensions can manifest as worry and guilt when individuals engage in restful activities, thereby highlighting the complex psychological mechanisms underlying rest intolerance in contemporary achievement-oriented societies.

High rest intolerance correlates with various negative psychological outcomes, including anxiety, depression, and reduced well-being [1,7]. Furthermore, individuals exhibiting high rest intolerance typically demonstrate decreased engagement in non-productive activities to mitigate self-directed guilt [2,8]. This poses a significant threat to university students' mental and physical health, potentially leading to long-term psychological health issues and a diminished quality of life.

Despite preliminary explorations of rest intolerance [2], research examining its heterogeneous patterns and formative mechanisms remains limited, particularly in non-Western cultural contexts. For instance, it remains unclear whether rest intolerance formation mechanisms occur in East Asian countries (e.g., China), where collectivistic values predominate, differs from their Western counterparts. While students in East Asian countries are commonly hypothesized to exhibit heightened rest intolerance due to their distinctive cultural value [3,8,9], insufficient empirical evidence currently precludes definitive conclusions.

This study aimed to investigate the influence of cultural value dimensions, specifically individualism and collectivism, and status anxiety on rest intolerance among university students within the Chinese cultural context. Clarifying the mechanisms through which cultural orientations affect rest intolerance is important for a deeper understanding of its origins and for creating campus environments conducive to mental health. This research may help university students balance study and rest, thereby promoting their overall psychological and physical well-being.

### 1.1. The potential relationship between individualism-collectivism and rest intolerance

In the Chinese context, cultural landscape of values presents a form of "hybrid modernity" [10] characterized by the coexistence of Confucian collectivism and modern competitive consciousness. This cultural configuration creates a unique

psychological tension for Chinese university students. While individual achievement is often framed as a way to honor the collective (e.g., bringing glory to one's family), students must still navigate the pressures of fulfilling collective obligations while engaging in intense individual competition [11,12]. This complex interplay, rather than a simple conflict, may intensify the experience of rest intolerance. Specifically, the high value placed on achievement (which serves both individualistic and collectivistic goals) can frame personal rest—often seen as a purely self-serving activity—as being incompatible with the constant need to strive, thereby fostering guilt and anxiety during leisure.

Hofstede's individualism-collectivism framework, when integrated with Triandis's vertical-horizontal dimensions [13], provides a robust theoretical lens for examining these behavioral tensions. The framework distinguishes between individualism (prioritizing the self over the group) and collectivism (prioritizing the group over the self), and further refines these with a vertical-horizontal dimension. The vertical dimension emphasizes hierarchy, status, and competition, whereas the horizontal dimension emphasizes equality and similarity. This creates four distinct cultural patterns: Horizontal Individualism (HI) reflects a desire for uniqueness and self-reliance without a strong drive to be superior to others. Vertical Individualism (VI) is characterized by a strong motivation for personal achievement, competition, and gaining status. Horizontal Collectivism (HC) involves an emphasis on interdependence, sociability, and common goals with others who are perceived as equals. Finally, Vertical Collectivism (VC) underscores the importance of group integrity, respecting authority within the group's hierarchy, and fulfilling one's duties to the in-group. As a prototypical vertical collectivist society, China emphasizes hierarchical structures and the primacy of collective objectives—reflected in traditional maxims such as "academic excellence leads to official appointment" and "all occupations are inferior, only scholarship is superior." Within higher education contexts, these cultural attributes manifest as intense grade-based competition and academic "involution" phenomena [14]. Furthermore, the cultural valorization of diligence (exemplified by proverbs like "heaven rewards the diligent") intersects with competitive pressures, potentially framing rest as a form of "moral transgression" that engenders heightened rest intolerance [3,15]. This differs conceptually from findings in Western contexts, where research suggests that rest intolerance emerges primarily from individualistic biases [2]. Given these cross-cultural disparities, further investigation is warranted to determine the relative influence of specific cultural value dimensions on rest intolerance experiences.

### 1.2. The mediating role of status anxiety in the relationship between individualism-collectivism and rest intolerance

The pathway from individualistic and collectivistic orientations to rest intolerance is likely not just direct, but also channeled through specific psychological states. Based on the principle of temporal and dispositional precedence, we propose that relatively stable cultural orientations, precede more transient affective-cognitive states like status anxiety. Therefore, we posit that status anxiety functions as a key mediating mechanism, channeling the effect of these cultural value dimensions onto individuals' experiences of rest [2,5,16,17]. Status anxiety refers to individuals' concerns and apprehensions about failing to meet socially defined standards of success, such as difficulty in advancement or prolonged career stagnation [5]. Research indicates that status anxiety functions as a chronic stressor with adverse effects on psychological well-being and subjective happiness [6], including but not limited to physical and mental health and overall life satisfaction [18].

Existing literature demonstrates that cultural values directly influence status anxiety [19]. Hofstede's four-dimensional theory of cultural values suggests that cultures characterized by high-power distance and high uncertainty avoidance are more conducive to status anxiety [20]. However, within the Chinese cultural context, the impact of these cultural dimensions on status anxiety may manifest distinctive characteristics: Chinese university students face not only institutional competition (e.g., national college entrance examinations, civil service examinations) but also horizontal comparisons and expectations from peers and across generations (e.g., social media "success narratives" and familial "social mobility" expectations). These multidimensional pressures contributed to chronically elevated status anxiety in this population.

The connection between status anxiety and rest intolerance demonstrates cultural embeddedness within this demographic group. As "involution" (excessive competition) becomes a defining characteristic of higher education, leisure is

perceived as sacrificing competitive advantage, thereby intensifying feelings of guilt. In a social atmosphere emphasizing "striving above all" and "not falling behind," university students frequently conflate personal worth with external achievements, such as academic performance and employment prospects. Research suggests that cultural environments that emphasize extrinsic values (e.g., wealth, social recognition, and reputation) increase individual levels of status anxiety [21]. Within the Chinese cultural context, the Confucian concept of the "face" is closely associated with status anxiety [11]. Specifically, in cultures that prioritize social harmony and collective honor, individuals are more susceptible to status anxiety stemming from concerns about losing face." Within this value system, leisure activities are negatively labeled as "lazy" or "unambitious," causing individuals to experience self-blame and guilt when resting, fearing being surpassed by peers or disappointing parental expectations—constituting the quintessential phenomenon of rest intolerance.

### 1.3. Current study

Based on the limitations of previous research, we aimed to investigate the influence of cultural values on rest intolerance by utilizing network analysis [22,23] to identify potential relationships among these constructs and to determine core symptoms. Furthermore, we employed mediation analysis to uncover the deeper mechanisms underlying the relationships between the variables.

Given the exploratory nature of our investigation, particularly the application of network analysis to identify the most salient cultural factors, we framed our inquiry around two primary research questions.

Q1: Among the four dimensions of cultural values (HI, VI, HC, VC), which dimension is most strongly associated with rest intolerance in Chinese university students?

Q2: Does status anxiety mediate the relationship between the key cultural value dimension identified in Q1 and rest intolerance? (Fig 1). presents the conceptual model guiding the exploration of the mediation pathway investigated in Q2.

## 2. Materials and methods

### 2.1. Ethics statement and participants

This study was approved by the Committee for Scientific Research and Academic Ethics of Anqing Normal University (AQNU2025001). Participants were recruited from a university in China via the Naodao Researcher Platform between March 17 and April 17, 2025. Prior to the study, all participants provided documented informed consent by confirming their agreement on the online platform. For one underage first-year student, consent was additionally obtained from their class counselor, who acted as their on-campus guardian. An attention check item ("Please select 'Strongly disagree'.") was embedded in the survey. Of the 613 initial recruits, 63 were excluded for failing this check. The final sample consisted of 550 participants (240 males, 310 females; $M_{age} = 21.05$, $SD_{age} = 1.56$).

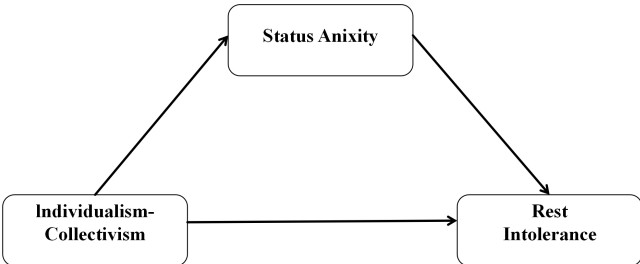

**Fig 1. Conceptual model for the mediation pathway (Q2).**

## 2.2. Instruments

**2.2.1. Demographic variables.** Participants reported their personal information, including their gender, age, major (Arts/Humanities, Science, Engineering) and current educational level (junior college, undergraduate, or postgraduate).

**2.2.2. Rest intolerance scale.** The 8-item RIS scale developed by [24] was used in this study. The scale contains four dimensions: Negative Feeling (2items; e.g., "When I am resting or having fun, I feel guilty."), Social Comparison (2items; e.g., "When I'm resting or having fun, I worry about being passed by someone else."), Obsessive Thinking (2items; e.g., "When I'm resting or having fun, I always feel that there's something else I'm not doing."), Cognitive Bias (2items; e.g., "I think I should need to spend more time studying and working rather than resting."). Participants used a Likert scale from 1 (strongly disagree) to 5 (strongly agree) to indicate their level of agreement with the following items. The authors of the scale have provided a corresponding Chinese version [25]. In this study, Cronbach's alpha coefficient was 0.869.

**2.2.3. Individualism-collectivism scale.** The Individualism-Collectivism Scale developed by [13] was used in this study. The scale consisted of four dimensions: Horizontal individualism (4items; e.g., "I'd rather depend on myself than others."), Vertical individualism(4items; e.g., "It is important that I do my job better than others."), Horizontal collectivism(4items; e.g., "If a coworker gets a prize, I would feel proud."), Vertical collectivism(4items; e.g., "Parents and children must stay together as much as possible."). The participants used a Likert scale ranging from 1 (Strongly Disagree) to 5 (Strongly Agree) to indicate their level of agreement with the extent to which they agreed with the items. The scale has been revised in Chinese, and has demonstrated good reliability in research studies [26]. In this study, the Cronbach's alpha coefficients for the four subscales were as follows: Horizontal Individualism ($\alpha = 0.641$), Vertical Individualism ($\alpha = 0.684$), Horizontal Collectivism ($\alpha = 0.752$), and Vertical Collectivism ($\alpha = 0.698$).

**2.2.4. Status anxiety questionnaire.** The Status Anxiety Questionnaire developed by [6] was used in this study. The questionnaire contains 5items (e.g., "I feel anxious that I will be stuck in my position for life."). Participants used a Likert scale ranging from 1 (Strongly Disagree) to 7 (Strongly Agree) to indicate their level of agreement with the following items. This scale has been used in previous research studies [2]. In this study, Cronbach's alpha coefficient was 0.864.

## 2.3. Data analysis

Data analysis was conducted using R (version 4.5.0) [27]. First, a network analysis was performed to explore the structural relationships among the constructs. The network was estimated on nine nodes: the four dimensions of the Individualism-Collectivism Scale (VI, HI, VC, HC), Status Anxiety (SA), and the four sub-dimensions of the Rest Intolerance Scale (RIS; i.e., negative feelings, social comparison, obsessive thinking, and cognitive bias).

Network estimation and visualization were implemented using the qgraph, igraph, and bootnet packages [28]. To control for spurious correlations, the network model was estimated using the graphical least absolute shrinkage and selection operator (GLASSO) with the extended Bayesian Information Criterion (EBIC) for regularization [23,29]. Furthermore, to ensure the robustness of the network structure, gender, age, and educational level were included as covariates. Therefore, the resulting graph is a partial correlation network, where edges represent the statistical relationships between nodes after controlling for these demographic variables. The network was visualized using the "spring" layout algorithm, with positive and negative associations represented by blue and red edges, respectively.

The stability of the network parameters was evaluated using 1,000 bootstrap iterations to estimate the confidence intervals of edge weights and the stability of centrality indices (CS-coefficient) [23,28]. To identify the nodes most critical in connecting the constructs, a bridge centrality analysis was conducted [30]. For this analysis, we defined three communities: "Individualism-Collectivism" (VI, HI, VC, HC), "Status Anxiety" (SA), and "Rest Intolerance" (the four RIS sub-dimensions). Finally, simple mediation analyses were conducted using Model 4 in PROCESS (v. 3.5) for SPSS 26.0 [31] to further examine specific pathways, with gender and age included as covariates. Mediation effects were considered significant if the 95% CI from 5,000 bootstrap samples did not include 0 [32].

## 3. Results

### 3.1. Common method bias test

To control for common method bias (CMB), this study used different measurement scales as research tools to mitigate CMB caused by self-reported single data sources [33]. To assess the presence of CMB, Harman's one-way test for common method bias was used, and the results showed that there were seven factors with eigenvalues greater than 1, and the first factor explained 22.632% of the variance, which is less than the critical criterion of 40%; therefore, the data of the present study did not have severe CMB.

### 3.2. Descriptive statistics and correlations

The demographic composition of the 550 participants was as follows. In terms of gender, there were 240 males (43.6%) and 310 females (56.4%). Regarding educational level, 17 participants were in junior college (3.1%), 482 were undergraduates (87.6%), and 51 were postgraduates (9.3%). The distribution of academic majors was: 175 in science (31.8%), 263 in arts/humanities (47.8%), and 112 in engineering (20.4%). Table 1 presents the descriptive statistics and correlations for all variables in Chinese college students. The assumption of normality for each variable was assessed prior to the preliminary analysis. The absolute values of skewness ranged from -0.627 to -0.146 and kurtosis ranged from -0.472 to 0.599. These results indicate that the variables satisfied the univariate normality assumption [34].

To explore these relationships further, we conducted an analysis to test the effect of these cultural value dimensions on the relationship between status anxiety and rest intolerance. As shown in the Table 1, rest intolerance (RIS) was significantly and positively correlated with vertical individualism (VI; $r = 0.52$, $p < 0.001$) and status anxiety (SA; $r = 0.41$, $p < 0.001$). Furthermore, vertical individualism and status anxiety were also strongly and positively correlated ($r = 0.49$, $p < 0.001$). These significant associations among the three core variables provided the empirical basis for the subsequent mediation analysis. Given that gender, age, and education were also significantly correlated with the study variables, they were included as covariates in all subsequent analyses to control for potential confounding effects.

### 3.3. Network analysis result

To elucidate the structural relationships among the dimensions of individualism-collectivism, status anxiety, and the components of rest intolerance, we estimated a regularized partial correlation network. The analysis was conducted on nine

**Table 1. Descriptive Statistics and Intercorrelations for Key Variables.**

| Varible | M | SD | 1 | 2 | 3 | 4 | 5 | 6 | 7 | 8 | 9 |
|---|---|---|---|---|---|---|---|---|---|---|---|
| 1.Gender[a] | 1.56 | 0.50 | – | | | | | | | | |
| 2.Age | 21.05 | 1.56 | −0.10* | – | | | | | | | |
| 3.Edu[b] | 2.06 | 0.35 | −0.08 | 0.42*** | – | | | | | | |
| 4.Major[c] | 1.89 | 0.71 | −.003 | −0.12* | 0.01 | – | | | | | |
| 5.HI | 16.06 | 2.20 | −0.27 | 0.17*** | 0.07 | 0.03 | – | | | | |
| 6.VI | 14.22 | 2.52 | −0.99* | 0.09* | 0.09* | −0.02 | 0.26*** | – | | | |
| 7.HC | 14.71 | 2.47 | −0.10* | 0.03 | 0.01 | −0.02 | 0.00 | −0.03 | – | | |
| 8.VC | 15.24 | 2.58 | −0.29*** | 0.07 | 0.01 | 0.00 | 0.09* | 0.19*** | 0.42*** | – | |
| 9.RIS | 25.16 | 6.45 | −0.79 | −0.05 | −0.03 | 0.03 | 0.12** | 0.52*** | −0.11** | 0.14** | – |
| 10.SA | 23.76 | 5.42 | −0.03 | 0.11* | 0.03 | 0.00 | 0.28*** | 0.49*** | −0.14** | 0.05 | 0.41*** |

Note: $N = 550$. M = mean, SD = standard deviations. HI = Horizontal individualism; VI = Vertical individualism; HC = Horizontal collectivism; VC = Vertical collectivism; RIS = Rest Intolerance Scale; SA = Status Anxiety. * $p < 0.05$, ** $p < 0.01$. *** $p < 0.001$.

[a]Gender, 1 = male, 2 = female, [b] Edu = educational level, 1 = Junior College, 2 = Undergraduate, 3 = Postgraduate,[c]Major, 1 = Arts/Humanities, 2 = Science, 3 = Engineering.

nodes, controlling for gender, age, and educational level. The final network model, selected via the Extended Bayesian Information Criterion (EBIC), consisted of 27 non-zero edges out of a possible 36, with a mean edge weight of 0.083. The overall structure of the network is depicted in Fig 2.

Several key edges emerged as particularly strong, providing initial answers to our research questions. Notably, there was a strong positive correlation between Vertical Individualism (VI) and Status Anxiety (SA; edge weight = 0.30). This represents the strongest link between the cultural values community and the other nodes. Furthermore, within the rest intolerance community, VI was most strongly connected to the Cognitive Bias (CB; edge weight = 0.18) dimension, while SA was most strongly connected to the Negative Feelings (NF; edge weight = 0.10) dimension. These specific connections highlight the primary pathways through which VI and SA are linked to the experience of rest intolerance.

**Network Stability and Accuracy.** The robustness of the estimated network was rigorously assessed through non-parametric bootstrapping with 1,000 samples [35]. The bootstrapped confidence intervals for individual edge weights were generally narrow, confirming the accuracy of the estimated edge parameters (S1 Fig). A case-dropping bootstrap procedure was employed to evaluate the stability of centrality indices. The results indicated excellent stability for our primary metric of interest, bridge strength, yielding a correlation stability coefficient (CS-Coefficient) of 0.595 (S2 Fig). This value is well above the recommended threshold of 0.5, suggesting that the centrality estimates are highly reliable and interpretable [35].

Furthermore, bootstrapped difference tests confirmed that the estimated edge weights and bridge centrality values were statistically distinguishable. The network's strongest edges were found to be significantly stronger than many weaker edges (S3 Fig), and the most central nodes, particularly VI and SA, had significantly higher bridge strength than other nodes in the network (S4 Fig). These tests collectively validate the robustness and interpretability of our findings.

**Node Predictability.** We examined the predictability of each node, quantified by its R-squared (R2) value, which represents the proportion of variance in a node explained by its neighbors. The predictability varied across the network, ranging from 0.12 for Horizontal Individualism (HI) to a high of 0.57 for Social Comparison (SC). Notably, the nodes within the Rest Intolerance community demonstrated substantial predictability (Negative Feelings, $R^2 = 0.56$; Social Comparison,

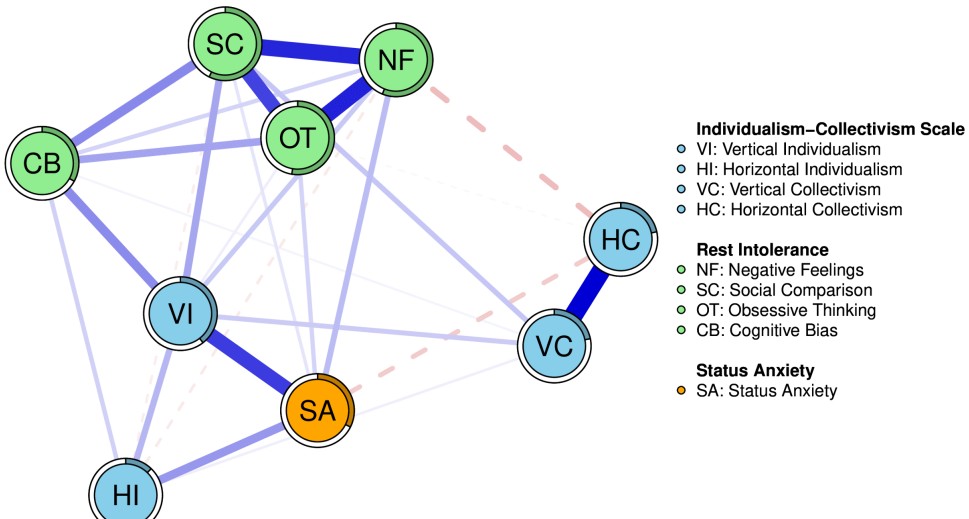

**Fig 2. The Network diagram of Individualism-Collectivism, Status Anxiety, and Rest Intolerance.** Note: In the network, each node represents a variable. Blue lines (edges) indicate positive partial correlations, while red lines indicate negative partial correlations. The thickness and saturation of an edge correspond to the strength of the correlation. The ring around each node represents its predictability (R2), with the same color area indicating the proportion of variance explained by its neighboring nodes.

$R^2 = 0.57$; Obsessive Thinking, $R^2 = 0.53$), indicating that these components are strongly interconnected within the psychological system.

**Bridge Centrality Analysis.** To address our first research question (Q1) and identify the key nodes connecting the conceptual communities, we conducted a bridge centrality analysis [30]. Three communities were defined: "Individualism-Collectivism Scale" (VI, HI, VC, HC), "Status Anxiety" (SA), and "Rest Intolerance" (NF, SC, OT, CB). Among the various bridge centrality indices, bridge strength is particularly informative as it quantifies the total strength of a node's connections to nodes in other communities.

The results are visualized in Fig 3, which highlights the identified core bridge nodes. Nodes colored in red (VI and SA) were identified as core bridge nodes, as they ranked within the top 20% of bridge strength values. Among these, the bridge centrality plot (Fig 4) revealed that Vertical Individualism (VI; bridge strength = 2.21) exhibited the highest bridge strength value among the four cultural value dimensions. This procedure confirmed that VI serves as the critical conduit linking cultural value orientations with the experience of rest intolerance, highlighting it as the most influential cultural dimension in connecting the different constructs of the network.

The network structure, particularly the identification of VI as a key bridge node connecting to the SA and Rest Intolerance communities, provides strong evidence for a potential pathway of influence. This finding directly informs our second research question and provides an empirical basis for formally testing the mediating role of status anxiety in the relationship between vertical individualism and rest intolerance.

**Supplementary Analysis.** To enhance the robustness of our findings, we further estimated an alternative network structure where the four dimensions of rest intolerance (NF, SC, OT, CB) were aggregated into a single total score (RIS). The resulting network is presented in S5 Fig.

### 3.4. Mediation effect analysis

The network analysis provided a structural map of relationships and identified VI and SA as critical bridge nodes. To formally test the most plausible directional pathway suggested by this structure and our theoretical framework, we next

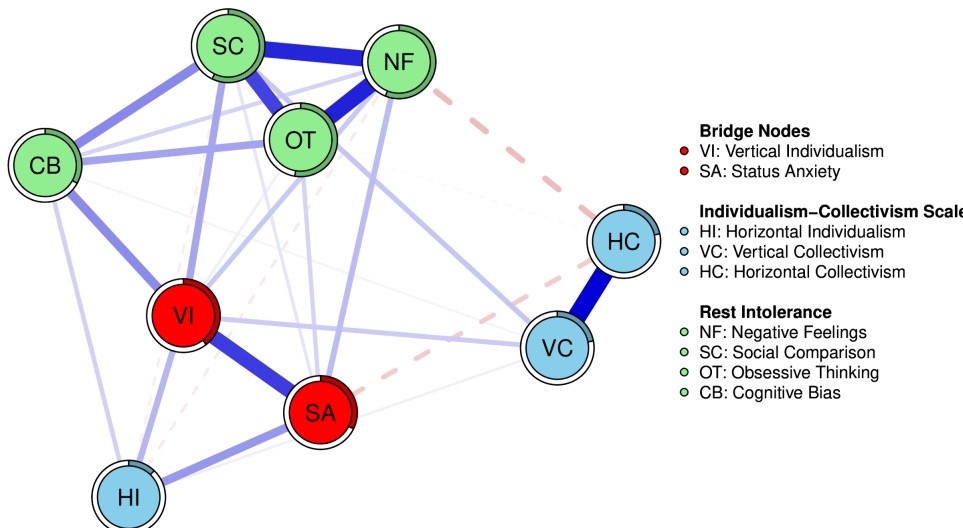

**Fig 3. Network Plot Highlighting Core Bridge Nodes.** Note: This figure highlights the identified core bridge nodes. Nodes colored in red (VI and SA) were identified as the core bridge nodes, defined as those in the top 20% of bridge strength values. These nodes are the most critical in connecting the different conceptual communities within the network.

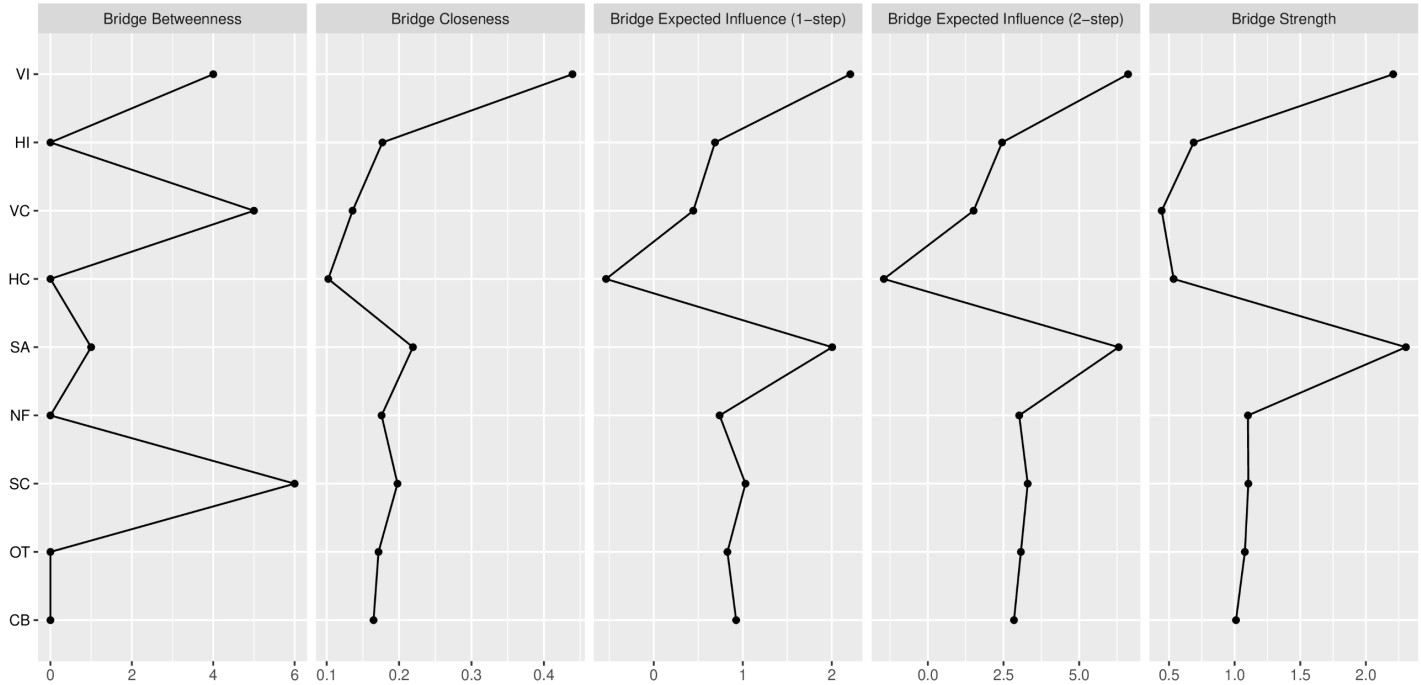

**Fig 4. Bridge Centrality Indices.**

conducted a simple mediation analysis. It is important to note that this cross-sectional mediation serves as a plausibility test for the proposed mechanism, rather than a definitive causal claim. We specify our primary theoretical model, VI→SA→RIS.

First, we tested the primary mediation model with the total score of Rest Intolerance (RIS) as the outcome variable, controlling for gender, age, and educational level. The results revealed that Vertical Individualism (VI) significantly predicted Status Anxiety (SA) ($b = 1.05$, $t = 13.04$, $p < 0.001$). In turn, SA significantly predicted RIS ($b = 0.25$, $t = 5.23$, $p < 0.001$). The direct effect of VI on RIS remained significant ($b = 1.08$, $t = 10.29$, $p < 0.001$). Crucially, the bootstrap analysis confirmed a significant completely standardized indirect effect of VI on RIS through SA ($\beta = 0.10$, 95% CI [0.05, 0.16]). This indirect effect accounted for 19.23% of the total effect (total effect $\beta = 0.52$), indicating that status anxiety partially mediates the relationship.

To further explore the nuances of this mechanism, we conducted four additional exploratory mediation analyses with each of the four sub-dimensions of rest intolerance as the outcome variable. The results revealed that the indirect effect of VI through SA was significant for Negative Feelings ($\beta = 0.12$, 95% CI [0.06, 0.18]), Social Comparison ($\beta = 0.09$, 95% CI [0.04, 0.14]), and Obsessive Thinking ($\beta = .11$, 95% CI [0.06, 0.16]). However, the indirect effect was not significant for Cognitive Bias ($\beta = 0.02$, 95% CI [−0.02, 0.08]). These findings suggest that the mediating pathway is most pronounced for the affective, social-comparative, and ruminative components of rest intolerance (Fig 5).

## 4. Discussion

The present study investigated the psychological mechanisms linking cultural values to rest intolerance among Chinese university students, employing a complementary two-pronged analytical approach. A nine-node partial correlation network analysis first revealed that Vertical Individualism (VI) and Status Anxiety (SA) were the most critical bridge nodes connecting the community of cultural values to the community of rest intolerance dimensions. A subsequent series of mediation

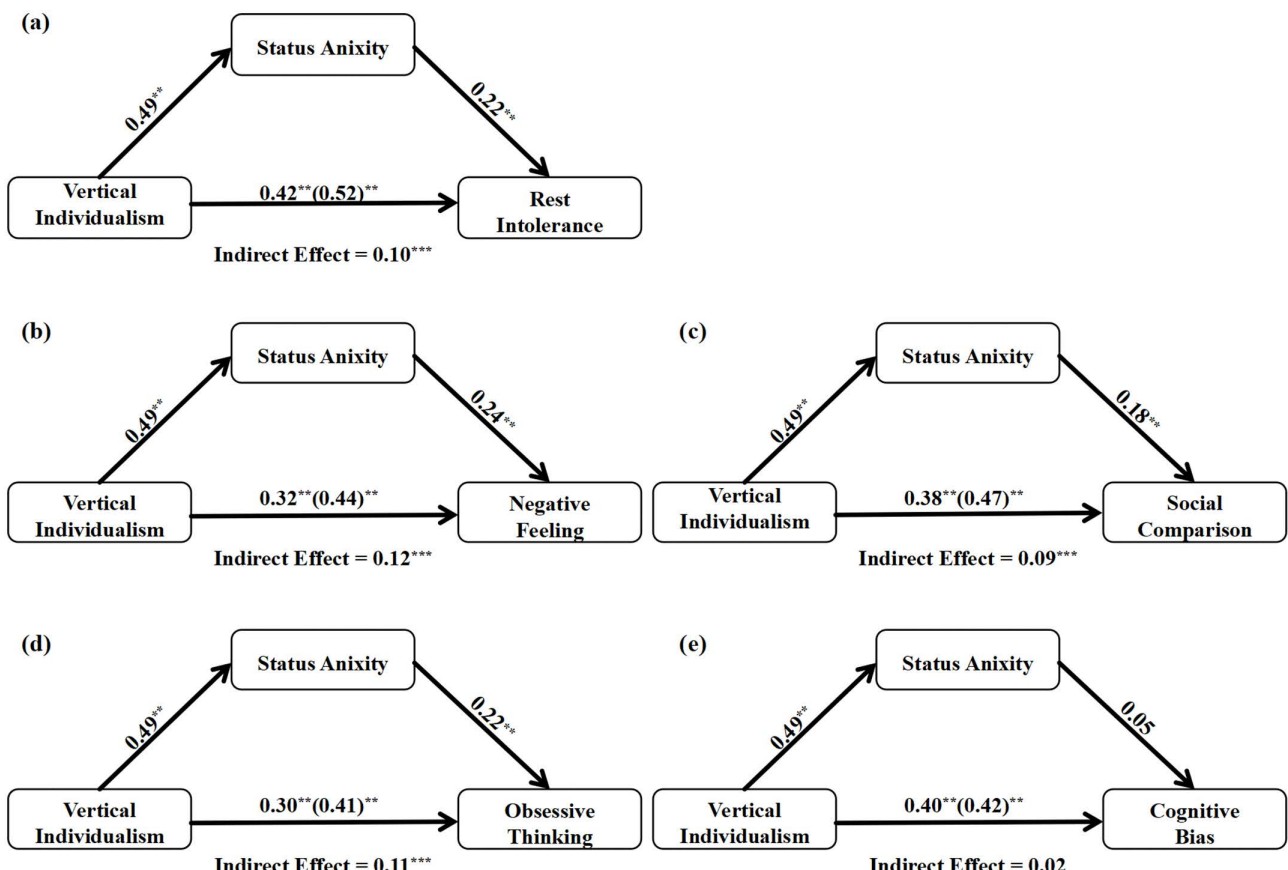

**Fig 5. Path diagrams for the mediation models.** Note: Subfigure (a) displays the primary mediation model with Vertical Individualism as the independent variable, Status Anxiety as the mediator, and the total score of Rest Intolerance as the dependent variable. Subfigures (b–e) display the exploratory mediation models for the four sub-dimensions: **(b)** Negative Feelings, **(c)** Social Comparison, **(d)** Obsessive Thinking, and **(e)** Cognitive Bias, respectively. Note: Path coefficients are completely standardized betas ($\beta$). The value on the path from the independent to the dependent variable represents the direct effect (c'), while the value in parentheses represents the total effect **(c)**. ** $p<0.001$.

analyses then formally tested this pathway. The results confirmed that SA partially mediated the relationship between VI and overall rest intolerance. Collectively, these findings challenge traditional collectivist explanations for this phenomenon and highlight the crucial role of competitive individualism in fostering rest intolerance in the contemporary Chinese context.

By modeling the sub-dimensions of cultural values and rest intolerance as distinct nodes, our network analysis moved beyond a general exploration to uncover a more detailed psychological architecture. The identification of VI as a key bridge node, particularly given its high bridge strength, provides a compelling, data-driven explanation for its influence. This suggests that it is the competitive, status-oriented facet of individualism, rather than a desire for uniqueness alone (Horizontal Individualism), that serves as the primary cultural gateway to experiencing discomfort with rest. This aligns with theories of social comparison [6], which posit that individuals who strongly identify with competition and status hierarchies are more likely to continuously activate status anxiety through upward comparisons. The network structure makes this pathway visible, illustrating how VI acts as a central hub connecting to both SA and various facets of rest intolerance.

The mediation analyses further clarified this pathway and added directional plausibility. The primary finding—that SA partially mediates the link between VI and overall rest intolerance—illuminates a core mechanism: individuals who endorse competitive values are more prone to status anxiety, which in turn amplifies their guilt during leisure. Our

exploratory analyses on the sub-dimensions provided even richer insights. The indirect effect was significant for the affective and ruminative components of rest intolerance (Negative Feelings, Social Comparison, Obsessive Thinking), but not for Cognitive Bias. This distinction is theoretically meaningful. It suggests that status anxiety primarily fuels the feeling of being behind (social comparison), the *guilt* associated with it (negative feelings), and the inability to switch off from work-related thoughts (obsessive thinking). The non-significant finding for cognitive bias (e.g., "I should be working") implies that while SA creates the emotional and ruminative distress of resting, the cold, cognitive belief that rest is inherently unproductive may be more directly inculcated by VI itself, a pathway supported by the significant direct effect in our mediation model. This aligns with cognitive fusion models, where individuals high in VI more readily fuse the concepts of "resting" and "failing" into a single cognitive framework [36].

Despite the novel insights, this study has several limitations that offer avenues for future research. First, and most critically, the cross-sectional design precludes causal inferences. While our mediation analysis provides results that are consistent with our proposed theoretical model (VI→SA→RIS), the data cannot establish the temporal precedence required for causality. The tested pathway is based on the strong theoretical assumption that stable cultural values precede more malleable psychological states. Future longitudinal or experimental studies are essential to formally test this causal chain and confirm the mediating role of status anxiety over time. Second, as the sample was limited to Chinese university students, the generalizability of these findings requires validation across different cultural contexts and age groups. Third, the internal consistency for the Horizontal Individualism ($a = 0.64$) and Vertical Individualism ($a = 0.68$) subscales was below the conventional 0.70 threshold. Although these values are sometimes observed in Chinese samples, future research would benefit from using instruments with more robust psychometric properties.

The partial mediation finding also suggests other psychological mechanisms are at play. One possible alternative mediator is performance-contingent self-worth [37,38], whereby individuals high in VI base their self-esteem directly on continuous productivity and achievement. For these individuals, rest is not seen as restorative but as a direct threat to their sense of value, thus generating guilt. Another potential mechanism could be the cognitive framing of time as an opportunity cost [12]. Influenced by competitive "hustle culture," these individuals may perceive any leisure time as a resource that is being wasted instead of being invested to gain a competitive edge, leading to anxiety and intolerance for rest.

In conclusion, this study challenges the prevailing narrative by identifying vertical individualism, not collectivism, as a key cultural antecedent of rest intolerance in a competitive, modernizing society [16]. By demonstrating the bridging role of vertical individualism in a network and testing the mediating pathway through status anxiety, our findings provide a more nuanced understanding of why winning the race can cost individuals their peace. From a practical perspective, these results suggest that interventions should not only target the management of status anxiety but also address the underlying competitive beliefs that frame rest as a personal failure [2]. Cognitive reappraisal interventions, such as those in Acceptance and Commitment Therapy, that help individuals redefine leisure as a form of self-care and a vital component of success, may be particularly beneficial for those high in vertical individualism [36].

## Supporting information

**S1 Fig. Accuracy of the edge-weights for the current network model.**
(TIF)

**S2 Fig. Stability of Bridge Strength Centrality Index.**
(TIF)

**S3 Fig. Bootstrapped Difference Test for Edge Weights.**
(TIF)

**S4 Fig. Bootstrapped Difference Test for Bridge Strength Centrality.**
(TIF)

**S5 Fig. The Network diagram of Individualism-Collectivism, Status Anxiety, and Rest Intolerance (total score).**
(TIF)

## Author contributions

**Conceptualization:** Wenyan Cheng, Ling Cheng.

**Data curation:** Zhichao Qian.

**Formal analysis:** Ling Cheng.

**Funding acquisition:** Haoyu Wang.

**Investigation:** Wenyan Cheng, Ling Cheng.

**Methodology:** Wenyan Cheng.

**Project administration:** Haoyu Wang.

**Supervision:** Haoyu Wang.

**Validation:** Wenyan Cheng, Ling Cheng, Haoyu Wang.

**Writing – original draft:** Wenyan Cheng.

**Writing – review & editing:** Zhichao Qian.

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
