## [Decision Letter · Decision Letter 0]

11 Aug 2025

PONE-D-25-26911When Winning Costs Your Peace: How Does Vertical Individualism Hijack Relaxation Capacity? Network Analysis and Mediation ModelsPLOS ONE?

Dear Dr. Wang,

Each of the four cultural dimensions needs some clear description in the Introduction, so readers can have a clearer understanding of each and more appreciation of the differences among them. I believe that this study is primarily exploratory in nature and the two stated hypotheses were not specific about the exact independent variable(s). As such, it’d be more appropriate to frame these statements as research questions rather than hypotheses. Then, for the rest of the manuscript, please make appropriate changes for all instances when a hypothesis is being mentioned.    Please provide sample items for each scale (or subscale), corresponding to the variables used in the analyses.For the cultural orientation scale, please report alpha for each of the four subscales.Descriptive statistics – For gender, reporting mean and SD do not seem appropriate (Figure 2a). Please report proportions. Moreover, for better understanding of your sample, please report descriptive statistics for other demographic variables (in addition to gender and age) that you have measured.  Section 3.2 (Descriptive statistics and correlations): This section first reports regression results (Beta and p-value) for VI predicting SA and VI predicting RIS. Then, it describes a “correlation line test” with some positive and negative correlations. From the description and Figure 2b (“Pearson correlation diagram”), I assume these are raw Pearson correlations. If so, please provide a clear Pearson correlation table with r and p-values for all pairs (some correlations are very difficult to read). In the main text, you could then describe *all* significant correlation results with r and p values (reporting regression results will then become unnecessary in this subsection about descriptive statistics and correlations as they will be redundant), and direct readers to the full table. If there is any misunderstanding on my part, please clarify and still provide relevant statistics whenever a result is described in words.     Section 3.3 (Network analysis result): Please provide relevant statistics whenever a result is described in words. Thus, in the sentence “…significant positive correlations between the "vertical individualism" dimension and both status anxiety and rest intolerance, with the bootstrapped confidence intervals of the edge weight parameters confirming the reliability of the current network structure (Appendix Fig. 1)”, it would be helpful to report the relevant statistics in the main text.Section 3.4 (Mediation effect analysis): Please also report the standardized coefficient (beta) for the indirect effect, in addition to the confidence interval. It mentions that the indirect effect accounts for “approximately 20% of the total effect”. It would be better to report the exact ratio of the indirect effect to the total effect without using “approximately”. For Figure 5a, it would be helpful to indicate in the caption what the number in brackets represents. You may also consider adding the statistics for the indirect effect in the figure to make it more complete. For the reverse mediation model, please report all relevant statistics, paralleling the results reported for the original mediation model. For Figure 5b, please make the same changes as Figure 5a.  Section 4.2 (Mediation model of vertical individualism, status anxiety, and rest intolerance): Repeating statistics that have been reported earlier is not necessary in the Discussion section.As a partial mediation was found, speculations around what other mechanisms, in addition to status anxiety, might explain the relationship between VI and rest intolerance could be discussed in the Discussion section.

plosone@plos.org . A rebuttal letter that responds to each point raised by the academic editor and the reviewers. You should upload this letter as a separate file labeled 'Response to Reviewers'.A marked-up copy of your manuscript that highlights changes made to the original version. You should upload this as a separate file labeled 'Revised Manuscript with Track Changes'.An unmarked version of your revised paper without tracked changes. You should upload this as a separate file labeled 'Manuscript'.

We look forward to receiving your revised manuscript.

Kind regards,

Andy H. Ng

Academic Editor

PLOS ONE

Journal Requirements:

2. Please provide additional details regarding participant consent. In the ethics statement in the Methods and online submission information, please ensure that you have specified what type of consent you obtained (for instance, written or verbal, and if verbal, how it was documented and witnessed). If your study included minors, state whether you obtained consent from parents or guardians.

4. Please note that funding information should not appear in any section or other areas of your manuscript. We will only publish funding information present in the Funding Statement section of the online submission form. Please remove any funding-related text from the manuscript.

“This research was funded by the Key Programs of Humanity and Social Science Research Project of Anhui Educational Committee, 2024AH052762”

6. We note that you have indicated that there are restrictions to data sharing for this study. PLOS only allows data to be available upon request if there are legal or ethical restrictions on sharing data publicly. For more information on unacceptable data access restrictions, please see http://journals.plos.org/plosone/s/data-availability#loc-unacceptable-data-access-restrictions.

7. Please ensure that you refer to Figure 1 in your text as, if accepted, production will need this reference to link the reader to the figure.

Reviewers' comments:

Reviewer's Responses to Questions

**Comments to the Author**

1. Is the manuscript technically sound, and do the data support the conclusions?

Reviewer #1: Yes

Reviewer #2: Partly

2. Has the statistical analysis been performed appropriately and rigorously?

Reviewer #1: Yes

Reviewer #2: I Don't Know

3. Have the authors made all data underlying the findings in their manuscript fully available?

Reviewer #1: Yes

Reviewer #2: Yes

4. Is the manuscript presented in an intelligible fashion and written in standard English?

Reviewer #1: Yes

Reviewer #2: Yes

Reviewer #1: This study employed samples from the Chinese university student population to investigate the psychological mechanisms linking cultural values to rest intolerance through status anxiety. The research question carries substantial theoretical and practical significance, particularly in China's highly competitive educational environment, making the examination of cultural and psychological factors contributing to rest intolerance particularly timely. The advantage of this study lies in its first exploration of the causes of rest intolerance in the Chinese environment, although only exploratory. However, the manuscript would benefit from addressing the following concerns:

1 In lines 67-71, the author states that "This conflict may significantly intensify the experience of rest intolerance." However, from my perspective, individual achievement itself can be viewed as a means of contributing to the collective. For instance, pursuing personal development goals appears compatible with fulfilling collective obligations, suggesting individualism and collectivism may not necessarily contradict each other.Moreover, while the author emphasizes this conflict's impact on rest intolerance, I question whether this is indeed the primary factor. In my view, the causes are likely multifaceted - including family upbringing, cultural environment, and even individualism itself. The conflict mentioned by the author doesn't appear to be a definitive causal factor.

2 The statement in line 87, "The relationship between cultural values and rest intolerance may be influenced by additional factors," clearly suggests a moderating effect rather than mediation as the author claims. When variable B affects the relationship between A and C, this conceptually represents moderation (a conditional effect), not mediation (an intervening mechanism). The author appears to have misapplied these fundamental statistical concepts in their interpretation.

3 In lines 157-158, it appears the author intended to refer to "horizontal individualism, vertical individualism, horizontal collectivism, and vertical collectivism" (rather than the terms currently presented in the text).

4 Regarding lines 214-215, please maintain consistent terminology for the dependent variable throughout the manuscript. While "rest intolerance" may be alternatively referred to as "rest shame" in Chinese academic contexts, the English version should use a single standardized term (preferably "rest intolerance") for reader comprehension. This inconsistency also appears in lines 217 and 219. We recommend conducting a thorough proofreading of the entire manuscript to ensure terminological consistency for this key construct.

5 Please revise the legend for Figure 3. The current inclusion of "Rest Intolerance" in the figure legend appears functionally unnecessary as it does not contribute meaningful explanatory value to the graphical presentation.

6 In line 217, the author states "a positive correlation between resting shame and status anxiety," implying a positive association between rest intolerance and status anxiety. However, in line 177, the author specifies that "Positive and negative associations are represented by blue and red edges, respectively." This creates a contradiction, as the figure depicts rest intolerance and status anxiety connected by a red edge (indicating a negative correlation), which is inconsistent with the textual description.

7 The phrasing in lines 218-219 – "a negative correlation between resting shame and status anxiety and horizontal collectivism" – is structurally ambiguous and may lead to misinterpretation. It remains unclear whether: 1)resting shame correlates negatively with both status anxiety and horizontal collectivism, or 2) resting shame and status anxiety collectively correlate negatively with horizontal collectivism.

8 The centrality analysis in Section 3.3 attempts to identify the most central node in the network, but unfortunately fails to achieve its intended purpose. Rather than examining global network centrality - which is inherently influenced by node selection bias - the analysis should focus on bridge centrality metrics between the rest intolerance and individualism-collectivism subnetworks, as these would provide more meaningful insights into the core research questions. The authors must further justify their node selection strategy, particularly explaining why rest intolerance was treated as a unitary construct rather than being operationalized through its constituent dimensions. A critical remaining question is how the network structure and centrality findings might differ if the multidimensional components of rest intolerance were modeled as separate nodes in the analysis.

9 Why not control for covariates in network analysis?

10 While the use of cross-sectional mediation in this exploratory study may be marginally acceptable, the proposed mediation pathways among individualism-collectivism, status anxiety, and rest intolerance hold limited substantive value. Given that cross-sectional mediation can produce statistically significant effects based solely on correlations—without establishing temporal precedence or causal direction—the reported findings require stronger theoretical justification.

11 As a non-specialist in cultural value research, I find it unclear whether individualism-collectivism can adequately serve as a representative measure of cultural values, or how precisely these two constructs differ and relate to each other conceptually. This distinction is particularly important as the manuscript appears to use these terms interchangeably without proper clarification, potentially conflating the specific dimension of individualism-collectivism (which focuses on self-group relations) with the broader, multidimensional construct of cultural values that encompasses various societal orientations beyond just this single aspect. The authors should explicitly address this conceptual overlap and justify their operationalization choice, particularly given that cultural values typically include additional fundamental dimensions beyond the individualism-collectivism spectrum.

12 In the first paragraph of the discussion, the author needs to summarize the basic results of this study.

While the manuscript currently has several issues that need addressing, I strongly encourage the authors to undertake further revisions, and I look forward to seeing these concerns adequately resolved in their next submission. Finally, I would like to emphasize that rest intolerance is indeed a highly significant issue worthy of in-depth investigation, particularly within the Chinese cultural context where it may present unique manifestations and implications that warrant special scholarly attention.

Reviewer #2: Rest intolerance is an interesting topic, but I found the writing style to be too dense. Perhaps the authors could try to simplify what they are trying to say. A more elaborate definition of vertical collectivism would be useful for example. In it’s present form it is hard to read.

**Do you want your identity to be public for this peer review?** For information about this choice, including consent withdrawal, please see our Privacy Policy

Reviewer #1: **Yes: ** Fei Wang

Reviewer #2: No

---

## [Author Response · Author response to Decision Letter 1]

20 Aug 2025

Dear Editor & Reviewers:

Thank you very much for your valuable comments and guidance. I have carefully revised the manuscript according to the PLOS ONE formatting requirements and uploaded the updated version. In addition, the information on title, authors, and affiliations has been provided as a separate file ([PONE-D-25-26911] title, author, affiliations information). All text related to funding has been removed, and the contribution of the project sponsor, Haoyu Wang, has been clearly specified in the Author Contributions section.

We are in the process of submitting an application to the database website to ensure that the data will be openly accessible. The citation of Figure 1 has also been updated in the main text with the DOI: https://doi.org/10.57760/sciencedb.psych.00638.

The detailed responses to the reviewers’ comments are provided in the file ([PONE-D-25-26911] response_to_comments), which correspond precisely to the revisions marked in the “Revised manuscript with track changes.”

I would like to sincerely thank you for your thoughtful editorial guidance and also express my deep gratitude to Reviewer #1 for their insightful and constructive comments, which have been extremely helpful in improving the quality of this manuscript.

Thank you again for your time and consideration.

Sincerely,

Wenyan Cheng

---

## [Decision Letter · Decision Letter 1]

22 Sep 2025

PONE-D-25-26911R1When Winning Costs Your Peace: How Does Vertical Individualism Hijack Relaxation Capacity? Network Analysis and Mediation ModelsPLOS ONE?

Dear Dr. Wang,

We look forward to receiving your revised manuscript.

Kind regards,

Andy H. Ng

Academic Editor

PLOS ONE

Journal Requirements:

**Additional Editor Comments:**

Thank your for addressing my comments raised in the previous round. Please respond to all comments raised by the reviewer in this round in your revised manuscript.

With regard to the reviewer's first comment about not being able to find the graph for bridge centrality, you wrote in the main text that “The analysis revealed that Status Anxiety (SA; bridge strength = 2.30) and Vertical Individualism (VI; bridge strength = 2.21) exhibited the highest bridge strength values in the network, as visualized in the bridge centrality plot in Fig 3.” This suggests that Fig. 3 was intended to be the bridge centrality graph. However, the figure title does not explicitly use the term “bridge centrality”, which may have led to the impression that the graph is missing. If my understanding is correct, I’d recommend revising the title, so that it clearly states “Bridge Centrality Plot”; for example, “Bridge Centrality Plot Highlighting Core Bridge Nodes.”

Reviewers' comments:

Reviewer's Responses to Questions

**Comments to the Author**

Reviewer #1: All comments have been addressed

2. Is the manuscript technically sound, and do the data support the conclusions?

Reviewer #1: Yes

3. Has the statistical analysis been performed appropriately and rigorously?

Reviewer #1: Yes

4. Have the authors made all data underlying the findings in their manuscript fully available?

Reviewer #1: Yes

5. Is the manuscript presented in an intelligible fashion and written in standard English?

Reviewer #1: Yes

Reviewer #1: The revised version from the author has largely addressed my concerns, though there are still some minor modifications needed for the current version. The specifics are as follows:

I did not seem to find the graph for bridge centrality in the manuscript. Please supplement it.

Since your research question is “Q1: Among the four dimensions of cultural values (HI, VI, HC, VC), which dimension is most strongly associated with rest intolerance in Chinese university students?”, there is no need to mention “SA” in the section on bridge centrality metrics. You only need to emphasize that VI has the highest value among all four dimensions.

Please include the network results from your first version—where rest intolerance was included as a node—as supplementary material.

Compared to using simple mediation analysis, I believe a better approach would be to construct a mediation model using edge weights within the network analysis, as this controls for other potential effects. That said, based on the current results, I expect such an analysis would also yield significant findings. Therefore, I do not insist that the author make this change.

**Do you want your identity to be public for this peer review?** For information about this choice, including consent withdrawal, please see our Privacy Policy

Reviewer #1: **Yes: ** Fei Wang

---

## [Author Response · Author response to Decision Letter 2]

23 Sep 2025

Dear Reviewer #1,

Thank you once again for your time and for the constructive comments that have helped us further improve our manuscript. We have carefully addressed each of your points. Our responses are detailed below.

Comment 1: I did not seem to find the graph for bridge centrality in the manuscript. Please supplement it.

Response: We sincerely apologize for the oversight. You and the Academic Editor are correct to note that our original presentation was unclear. We have now supplemented the manuscript with a new, dedicated figure (Figure 4, titled "Bridge Centrality Indices") that specifically visualizes the bridge strength indices. The text has been revised to accurately reference this new figure when discussing the bridge centrality results.

Comment 2: Since your research question is “Q1: Among the four dimensions of cultural values (HI, VI, HC, VC), which dimension is most strongly associated with rest intolerance in Chinese university students?”, there is no need to mention “SA” in the section on bridge centrality metrics. You only need to emphasize that VI has the highest value among all four dimensions.

Response: Thank you for this precise suggestion. We agree that focusing the narrative directly on the research question improves clarity. In the revised manuscript, we have modified the relevant sentence in the Results section to emphasize that Vertical Individualism (VI) demonstrated the highest bridge strength among the four cultural dimensions, thereby providing a direct answer to Q1.

Comment 3: Please include the network results from your first version—where rest intolerance was included as a node—as supplementary material.

Response: As requested, we have included the network analysis from the initial version (featuring rest intolerance as a single node) as Supplementary Figure S5 in the supplementary materials. A reference to this figure has been added to the main text.

Comment 4: Compared to using simple mediation analysis, I believe a better approach would be to construct a mediation model using edge weights within the network analysis, as this controls for other potential effects. That said, based on the current results, I expect such an analysis would also yield significant findings. Therefore, I do not insist that the author make this change.

Response: We are deeply grateful for this insightful methodological suggestion and for your understanding. We acknowledge that a mediation model based on edge weights is a sophisticated approach. However, due to our current technical limitations with this specific method, we have decided to retain the original mediation analysis to ensure the reliability of our findings. We sincerely hope this is acceptable and thank you again for your expert advice and flexibility.

We believe the manuscript has been strengthened thanks to your input. Thank you for your continued guidance.

Sincerely,

Wenyan Cheng

---

## [Decision Letter · Decision Letter 2]

13 Oct 2025

PONE-D-25-26911R2When Winning Costs Your Peace: How Does Vertical Individualism Hijack Relaxation Capacity? Network Analysis and Mediation ModelsPLOS ONE?

Dear Dr. Wang,

We look forward to receiving your revised manuscript.

Kind regards,

Andy H. Ng

Academic Editor

PLOS ONE

Journal Requirements:

**Additional Editor Comments:**

 I noticed that the new Figure 4 displays the bridge strength values ranging from approximately 0.45 to 1.05. For Vertical Individualism in particular, the figure shows a value of about 0.9, whereas the text reports a value of 2.21 (lines 298 and 366). If my understanding is correct, could you please clarify this difference? For reader comprehension, please adjust either the figure or the text to ensure that they are consistent.

Reviewers' comments:

Reviewer's Responses to Questions

**Comments to the Author**

Reviewer #1: All comments have been addressed

2. Is the manuscript technically sound, and do the data support the conclusions?

Reviewer #1: Yes

3. Has the statistical analysis been performed appropriately and rigorously?

Reviewer #1: Yes

4. Have the authors made all data underlying the findings in their manuscript fully available?

Reviewer #1: Yes

5. Is the manuscript presented in an intelligible fashion and written in standard English?

Reviewer #1: Yes

Reviewer #1: (No Response)

**Do you want your identity to be public for this peer review?** For information about this choice, including consent withdrawal, please see our Privacy Policy

Reviewer #1: **Yes: ** Fei Wang

---

## [Author Response · Author response to Decision Letter 3]

14 Oct 2025

To Reviewer:

Thank you very much for your careful observation and constructive comment regarding the inconsistency between the value of Vertical Individualism (VI) reported in the text and the one displayed in Figure 4.

You are absolutely right — the previously submitted figure was mistakenly generated from the overall network Centrality Analysis rather than the Bridge Centrality Analysis. As a result, the bridge strength value for VI appeared incorrectly as approximately 0.9, while the text reported the correct value of 2.21.

We have now corrected this issue by revising our R code and regenerating Figure 4 using the proper Bridge Centrality Analysis procedure. The updated figure now accurately displays the bridge strength value of 2.21 for Vertical Individualism, ensuring consistency between the figure and the text.

To maintain the conciseness of the Discussion section, we have removed the original Strength value (2.21) at line 369 in the main text and replaced Fig. 4 accordingly.

We sincerely appreciate your attention to detail, which helped us identify and correct this oversight. Thank you again for your valuable feedback that improved the accuracy and clarity of our manuscript.

---

## [Editor Report · Decision Letter 3]

17 Oct 2025

When Winning Costs Your Peace: How Does Vertical Individualism Hijack Relaxation Capacity? Network Analysis and Mediation Models

PONE-D-25-26911R3

Dear Dr. Wang,

We’re pleased to inform you that your manuscript has been judged scientifically suitable for publication and will be formally accepted for publication once it meets all outstanding technical requirements.

Kind regards,

Andy H. Ng

Academic Editor

PLOS ONE
---

## [Editor Report · Acceptance letter]

PONE-D-25-26911R3

PLOS ONE

Dear Dr. Wang,

I'm pleased to inform you that your manuscript has been deemed suitable for publication in PLOS ONE. Congratulations! Your manuscript is now being handed over to our production team.

Kind regards,

on behalf of

Dr. Andy H. Ng

Academic Editor

PLOS ONE